# Impact of agile intuition on innovation behavior: Chinese evidence and a new proposal

Qing Zhao[1], Langang Feng[2,3], Hai Liu[1], Mei Yu[1]*, Shu Shang[1], Yuqi Zhu[1], YanPing Xie[1], Jing Li[1], Yuzhu Meng[1]

**1** College of Business Administration, Guizhou University of Finance and Economics, Guiyang, China, **2** High-End Think Tank of Guizhou Green Development Strategy, Guizhou University of Finance and Economics, Guiyang, China, **3** Guizhou Key Laboratory of Big Data Statistical Analysis, Guizhou University of Finance and Economics, Guiyang, China

* YuMeiYuMeiwork@126.com

**Data Availability Statement:** All relevant data are within the paper and its Supporting information files.

**Funding:** Langang Feng received the project. This research was funded by the National Social Science

## Abstract

With the tendency toward economic and strategy decoupling between China and the United States and amidst the anti-globalization trend, enterprises are facing unprecedented challenges and opportunities. In this study, we reveal how the agile intuition (AI) of top managers with respect to the external environment affects enterprise innovation behavior (IB) based on the cognition–behavior framework. Strategic learning (SL) is considered a moderator, and knowledge sharing (KS) is considered a mediator. The survey sample consists of 305 managers from 47 enterprises in China during the COVID-19 period. The empirical results show that top management agile intuition significantly promotes enterprise IB; knowledge sharing (KS) partially mediates the relationship between top manager AI and enterprise IB; and SL suppresses the promotion effect of top manager AI on enterprise IB to a certain extent, hindering blind innovation. In a surprising result, we find that strategic guidance by an external consultant does not significantly affect the enterprise IB in China.

## Introduction

Decoupling and the trend toward anti-globalization in the economic and strategic fields of China and the United States may damage the economic links between major countries [1] and the worldwide industrial chain. At the same time, under the influence of the COVID-19 period, China's economy experienced a hard time in the first quarter of 2020. However, statistical results from McKinsey & Company in June 2020 revealed that certain enterprises in China were not negatively impacted but, rather, exponentially increased their sales while gaining thousands of new customers, primarily in the areas of big data (e.g., B2B, B2C, Teleworking), mobile games (e.g., Tencent), personal consumption (e.g., O2O, Express-deliver/ Takeaway), artificial intelligence (in the technology-intensive field), public health but also in other fields. It can be noted that the process of significant change in the external environment and the construction of a new order are "dual character." Simultaneously, agile intuition of the

Foundation of China (grant number: 17BJY076) and Guizhou Province Philosophy and Social Science Planning Project, China (grant number: 21GZYB61). The funders had no role in study design, data collection and analysis, decision to publish, or preparation of the manuscript.

**Competing interests:** The authors have declared that no competing interests exist.

market by top managers, fast and efficient decision-making, and innovative behavior by organizations have been of central importance in riding out the crisis and related difficulties. The view presented here is similar to the Organizational Decision Procedure Theory proposed by Papadakis *et al.* [2]. In recent years, the environmental acuity of entrepreneurs has attracted the attention of scholars in different fields. for example, family business [3], service industry [4], interpersonal network under crisis [5], and family medical subsidy market [6].

Different from the view of traditional theories of strategy and marketing, environmental acuity is an essential component of an enterprise's knowledge capital, which refers to the speed with which an enterprise responds to changes in the market, policies, economics and other external environmental factors (under dynamic and uncertain conditions) [5]. The concept of intuition refers to people's sensitivity and ability to anticipate [7]. Thus, agile intuition can be characterized as a high degree of intuition and anticipatory ability with respect to changes in the external environment. Successful practices indicate that agile intuition on the part of managers (particularly top managers or entrepreneurs) can identify and predict external threats or opportunities and help an enterprise effectively avoid a severe enterprise crisis. In addition, managers who possess AI can transform crises into opportunities and create so-called business miracles [2]. A prominent early finding of the dual-system theory was that external information can trigger intuitive associations [8]. After assessing their own experience and knowledge, individuals can form sensitive/intuitive perceptions. However, such sensitive perceptions and knowledge can easily lead to biased decision-making [9]. Therefore, the rational processing system of information, another type of decision-support system, helps people correct agile perceptions and decisions and establish an accurate decision pattern [10].

Lian observed that the pressure of intensified competition and market reform is one of the forces that trigger enterprise innovation [11]. According to original proposition, entrepreneurs are the leading driving force for innovation [12]. As such, their agile intuition, which involves judgment and foresight regarding future environmental trends as well as policy, technology, and economic trends, constitutes the source of innovation. It can be inferred there is an influence relationship between manager intuition and enterprise innovation (here, the focus is on innovation behavior). From a theoretical perspective, dual-system theory has investigated the influence mechanism with respect to accurate information processing and knowledge spillover to a limited extent. Petty and Cacioppo noted that the perception of individuals should first be converted into systematic knowledge (i.e., made logical) [10] and then transmitted to the other members of the organization [13]. Finally, all organization members should pursue product or service innovation through knowledge digestion and relevant behavior. For example, Ren Zhengfei (The owner of HUAWEI Technologies Co., Ltd.) empirically predicted the current competitive environment of decoupling and anti-globalization between China and the US. By implementing an alternative innovation plan long in advance of this change, his team (HUAWEI) established the foundation for their firm's current remarkable success, which has been based on organizational innovation behavior.

Theoretically, it is a universal organizational-information perception and behavioral law that a top manager's agile intuition with respect to the environment is processed and transformed into the IB action path of organizational members. However, Dijksterhuis proposed that the processing and transmission of information/knowledge are costly in terms of time and resources [14]. He argued that decisions based on intuition were of higher quality when the decision makers faced complexity. Dijksterhuis's theory has been accompanied by substantial controversy and widely cited. It can be observed that this theory breaks with previous views regarding the conventional environmental background and undoubtedly provides a new way of thinking for rapid and high-quality decision-making when enterprises face significant change, adversity, and a highly complex environment. What factors can improve/inhibit

enterprise information processing and knowledge transfer and ultimately result in efficient behavior? Such questions represent the focus of our efforts. We believe that based on the high-echelons theory and leader-member exchange theory, most studies can prove that when managers become more sensitive to market changes, it can be translated into innovation activities throughout the enterprise so that they can gain more advantage in competition. Nevertheless, it is not yet clear how leaders communicate and share information and knowledge with corporate members during the transformation process, as well as the role of these important factors in the corporate strategy learning and knowledge accumulation of the entire organization. In addition, currently few related studies were found taking into account the mechanism of knowledge sharing and strategic learning.

Thus, this study converges the process of cognition-behavior with the operational procedure of enterprise strategy. Does cognition-behavior focus on investigating whether the agile intuition of top managers is conducive to organizational IB? If so, what is the mechanism of this link? In addition, what factors inhibit or promote such a link? Against this backdrop, we discuss the influence of top manager agile intuition (AI) on enterprise innovation behavior (BI)and confirm the mediating role of knowledge sharing (KS) and the moderating effect of strategic learning (SL).

## Literature review and hypothesis

### Effect of top management AI on enterprise IB

In the literature, there are three research paradigms for investigating the relationship between top manager AI and organizational IB: the strategic resource paradigm, the enterprise capability paradigm, and the perceived response paradigm. As previously mentioned, top management AI reflects the enterprise's anticipation of and response speed to environmental change, which implies two crucial characteristics of top management AI: timeliness and innovation. Sambamurthy et al. observed that a top manager's AI with respect to the environment is a key strategic resource, helping enterprises quickly adapt to a dynamic and changeable business environment [15]. Swafford et al. believe that the top manager AI enables enterprises to effectively perceive and respond to market changes and opportunities and promotes technological innovation [16]. Tallon and Pinsonneault found that the AI of top managers could help such managers identify changing market trends and then help them expand enterprise IB in the form of new products, services, or businesses while improving the technological innovation performance in a rapid response to changes [17]. Chakravarty et al. argued that AI is embodied in two crucial abilities [18]: 1) the exploration ability, which is manifested in the innovative actions undertaken by enterprises to obtain new competitive advantages and break the industry pattern (such as developing new products, services, and business models), and 2) the timely response and adjustment ability, which is manifested in the external environment (such as industry environment, consumer preference, production technology). Roberts and Grover believe that top manager AI includes two components: perception and response [19]. Perception involves the judgment and prediction vis-à-vis the market, including capturing consumer preferences and predicting competitor behavior and industry development trends. Response involves the enterprise's executive ability, that is, the capacity to develop and implement a response plan based on market intelligence. Harraf et al. noted that the thinking of enterprise top managers could be considered their intuition [20].

In terms of empirical research, Inman *et al.* analyzed 96 manufacturing enterprises and found that agile perception of the market by top managers could effectively improve enterprise financial, operation, and market performance [21]. Tallon and Pinsonneault analyzed survey data from 241 enterprises and found a positive correlation between the sensitivity of top

managers to the market and the financial performance of their firms (in terms of return on assets, net profit, operating income, and asset ratio) [17]. Chen *et al.* analyzed data on Chinese manufacturing enterprises [22]. They concluded that the sensitivity of top managers is conducive to the full use of market opportunities by these enterprises and the promotion of a series of competitive behaviors, including innovation. Based on this understanding, this paper proposes the following hypothesis:

**Hypothesis 1. (H1)**: Top manager AI regarding the market positively affects enterprise IB.

## Mediating role of KS

Regarding KS, Yan and Chen found four main influencing factors: type of products and services, nature of employees, nature of products, and company size [23]. Lin and Lee found that executive attitude, individual perceived behavior control, and subjective norms positively affect KS motivation [24]. Bock and Kim introduced social psychological factors and organizational culture factors and argued that rational behavior and external motivation would affect individual KS motivation [25]. Kim and Ju found that perception and reward mechanisms had an important impact on KS behavior in an empirical study, and there was no significant correlation between trust, cooperation, communication channels, and KS [26]. Enno proposed hypotheses and verified the significant positive effect of psychological security on KS behavior from the perspective of psychology [27].

To our knowledge, previous researchers have primarily regarded KS as a crucial factor influencing enterprise IB. For example, Eriksson and Dielcson observed that KS was a process in which people share knowledge and that in this process, subsequently, new knowledge is created and new capabilities formed [28]. Alavi and Leidner argued that KS is a dynamic process that diffuses everywhere [29]. Through KS, the knowledge an organization requires can flow between individuals and other organizations and consequently be carried in different channels, which has a decisive role in promoting OP. Bart and Ridder divided knowledge into explicit and tacit knowledge and found that KS is the process of mutual transformation between these knowledge forms and that from this process IB would result [30].

This study adopts KS as a mediator and investigates the practical means of information transmission after top managers recognize a market change trend through their AI. In the practice of enterprise management, both explicit knowledge and tacit knowledge can be shared through various channels. On the one hand, such KS can help an enterprise avoid the lag caused by the complicated process of establishing formal information transmission channels and achieve the goal of improved efficiency. On the other, it can form new knowledge in the course of sharing, thus providing an essential basis for enterprise innovation. Therefore, hypotheses H2, H3, and H4 are proposed as follows:

**Hypothesis 2. (H2)**: Top manager AI has a positive effect on KS.

**Hypothesis 3. (H3)**: KS has a positive effect on enterprise IB.

**Hypothesis 4. (H4)**: KS plays a mediating role in the positive effect of top manager AI on enterprise IB.

## Moderating role of SL

According to strategic learning theory, SL integrates learning from different levels of organizations and raises it to the status of strategy. Kuwada has long deemed that SL is a learning behavior and process that can improve enterprise long-term adaptability [31]. As an ecological

process within an organization, SL, including the process of strategic knowledge creation and strategic knowledge refinement, helps enterprises promote BI from a strategic perspective and represents long-term and breakthrough innovation. Helfat and Peteraf observed that SL could integrate organizational learning, focusing on knowledge acquisition and knowledge management and emphasizing knowledge application, which constitutes SL's core dimension [32].

Numerous studies have shown that SL can promote the ability of enterprises to absorb and process external information and knowledge and thus significantly impact enterprise IB. For example, Thomas *et al.* proposed that SL involves generating learning activities that can support future strategies and then shape knowledge heterogeneity that is conducive to organizational performance improvement [33]. Their research implies a moderating role for SL. Lu started with an analysis of the theoretical context and concept connotation of SL, noted the nature of the influence of SL on enterprise IB, and argued that the two core dimensions of SL (organizational learning and knowledge management) exerted a collective impact on IB [34]. Their research primarily focused on the direct effect of SL on IB, while knowledge integration (including the absorption and processing of market information) and SL and its impact were less discussed. Overall, the current literature on the direct impact of SL on IB is relatively informative. However, SL's moderating role is rarely discussed. In this study, we seek to verify this role. It can be noted that if SL is assumed to be a moderator, it will be helpful to further investigate the role of top manager AI in promoting or inhibiting enterprise IB and building a bridge between them. Therefore, we propose hypothesis H5a as follows:

**Hypothesis 5a. (H5a)**: SL moderates the positive effect of top manager AI on enterprise IB.

To investigate the further moderating role of SL, this study extends the mediated-moderating and moderated-mediating models and proposes hypotheses H5b and H5c:

**Hypothesis 5b. (H5b)**: SL mediates the facilitation of top manager AI on KS and thus affects enterprise IB.

**Hypothesis 5c. (H5c)**: KS is mediated by SL, which affects the influence of top manager AI on enterprise IB.

Based on the preceding discussion, we construct the study's theoretical model (Fig 1).

## Method

### Sample and data collection

First, in an experiment, we analyzed the current industry classification types and characteristics of Chinese enterprises using the more widely recognized classification methods, whereby we screened for industry type: manufacturing, trade, finance, traditional services, construction, and other (mainly mixed type industries). In each industry, we selected 5–10 representatives who were familiar with the research. After consent was obtained, the administrative department of each enterprise was entrusted with distributing a questionnaire. Finally, completed questionnaires from 47 enterprises, including from enterprises involved in manufacturing, trade, finance, and engineering, were collected. These enterprises are outstanding representatives in their respective industries. Among the enterprises, most of the manufacturing and trading enterprises are key, large state-owned enterprises. The surveyed financial, construction engineering and other enterprises are primarily enterprises with strong competitiveness in their industries, and they would be expected to shoulder the main burden of a Chinese business reaction if a sudden change in the international political and economic pattern were to occur. As the prominent leaders of these enterprises have worked or cooperated with the

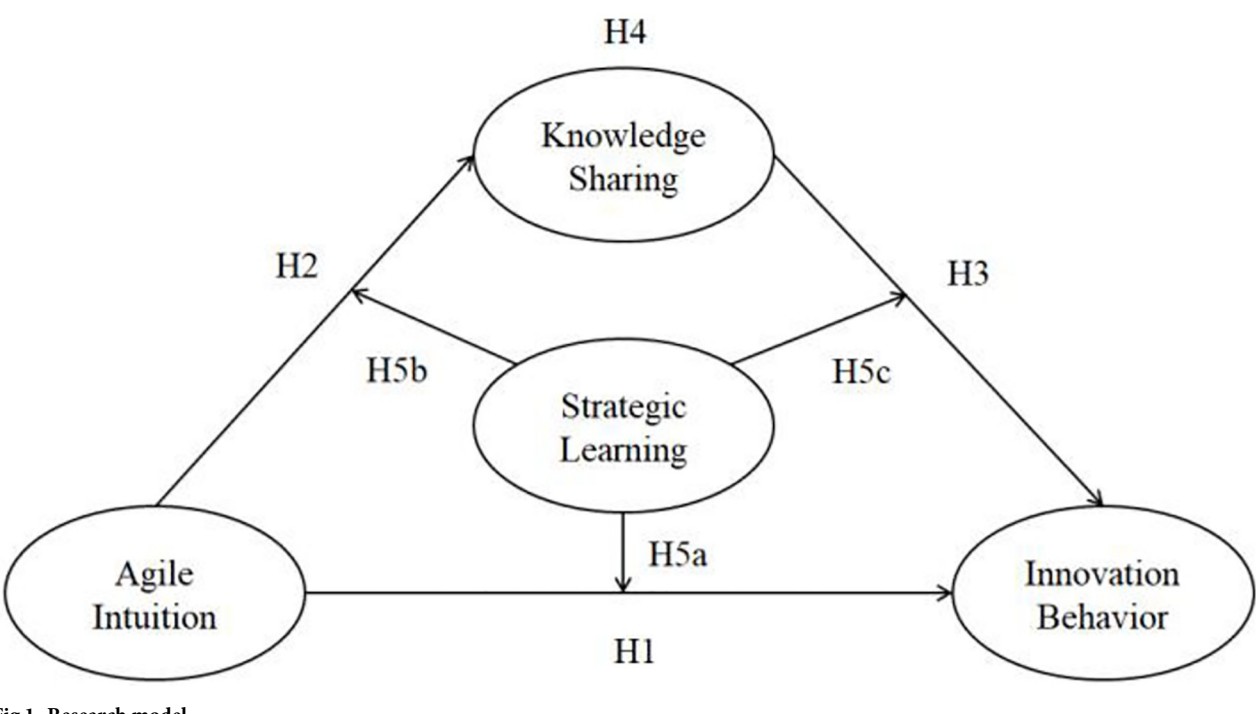

**Fig 1. Research model.**

research team for a long time, they viewed the survey seriously, thus ensuring the research's quality.

The questionnaire consisted of two parts: key information regarding the sample interviewees and a company profile, including gender, age, education level, position and working years, company size, and other company properties. In the central part of the questionnaire, we investigated items such as AI, KS, SL, and IB using a Likert 7-point positive scoring method. We conducted a total of two questionnaire surveys. Two sets of questionnaires were issued for the first time to measure AI and SL. The subjects included 112 senior executives from the parent company and 356 senior executives from the subsidiary. 936 copies distributed and 766 valid ones returned, with an effective recovery rate of 82.3%. The first test is SL. Three sets of questionnaires were issued for the second time to measure SL, KS and IB. The subjects were 468 non-leaders from the 47 enterprises, a total of 1404 copies distributed and 852 valid ones returned, with an effective recovery rate of 60.7%. As for the SL in both of the two surveys, we used exactly the same questionnaire in terms of participants, with slightly different expression of the questions, allowing the interviewees to accurately identify these questions. Totally 1618 copies of all the five sets of valid questionnaires were matched, and finally 305 data tables were obtained, which can completely cover the five latent variables.

A description of the personnel included in the survey sample is provided in Table 1. Men accounted for 47.39%, and women accounted for 52.61%. A total of 55.56% were aged 31 to 50 years, 78.03% had a university education or above, 42.6% were ordinary staff, and 57.4% were middle and senior managers.

The company position level of the managers is shown in Table 2. This group involves a wide range of individuals and has representativeness. The responding company types include the manufacturing, financial, traditional service, and construction engineering industries. Among them, the manufacturing industry and financial industry accounted for 12.46% and

**Table 1. Interviewers in the survey sample.**

| Gender | Proportion | Age | Proportion | Education | Proportion | Position | Proportion |
|--------|------------|-----|------------|-----------|------------|----------|------------|
| male | 47.39% | ≤30 | 24.84% | Primary school and below | 0.33% | Front line Management | 54.43% |
| | | 31–40 | 25.82% | Junior middle school | 3.28% | Middle manager | 22.62% |
| female | 52.61% | 41–50 | 29.74% | High school | 18.36% | Top managers | 2.95% |
| | | 51–60 | 17.97% | University | 57.70% | Ordinary staff | 20.00% |
| | | ≥60 | 1.31% | Master degree or above | 20.33% | | |

18.03% of the total number of responding managers, respectively. A total of 77.38% of the companies had been established for ten years or more, 40.00% of the companies had fewer than 1000 employees, and 38.03% had more than 3000 employees. The company's ownership type was mainly state-owned and state-controlled, accounting for 20.00% and 30.82%, respectively. Additionally, among the sample enterprises, 23.28% were high-tech enterprises, and 52.79% had received strategic guidance from third-party consulting companies.

**Table 2. The situation of sample enterprises.**

| Variables | Classification | Number of samples | Percentage (%) |
|-----------|----------------|-------------------|----------------|
| Industry Type | Manufacture | 38 | 12.46% |
| | Trade | 11 | 3.61% |
| | finance | 55 | 18.03% |
| | Traditional services | 18 | 5.90% |
| | Construction | 29 | 9.51% |
| | Others | 144 | 47.21% |
| Location | East | 52 | 17.05% |
| | Central section | 86 | 28.20% |
| | West | 166 | 54.43% |
| | Overseas | 1 | 0.33% |
| Year | ≤1 year | 7 | 2.30% |
| | 1–5 years | 18 | 5.90% |
| | 5–10 years | 44 | 14.43% |
| | ≥10years | 236 | 77.38% |
| Size | Less than 1000 people | 122 | 40.00% |
| | 1000–2000 | 52 | 17.05% |
| | 2000–3000 | 15 | 4.92% |
| | More than 3000 people | 116 | 38.03% |
| Properties | State owned | 61 | 20.00% |
| | State-controlled | 94 | 30.82% |
| | Private- controlled | 21 | 6.89% |
| | Private owned | 33 | 10.82% |
| | Foreign enterprise | 8 | 2.62% |
| | Others | 88 | 28.85% |
| Is it a high-tech enterprise | Yes | 71 | 23.28% |
| | No | 234 | 76.72% |
| Is it received strategic guidance | Yes | 161 | 52.79% |
| | No | 144 | 47.21% |

## Measurement

The measurement items were designed through three steps. First, given the relatively mature scale of the surveyed firms, the items were optimized and improved through a visiting survey of large domestic enterprises, and a preliminary design of the questionnaire was completed. Second, experts were consulted to improve the wording of the questionnaire so that it conformed to the enterprise management system and to ensure the respondents would fully understand the items. Third, the questionnaire content was further developed according to field research interviews. Thirty-two items were scored using a seven-point Likert scale (1 for total disagreement; 7 for full agreement).

The dependent variable is enterprise IB. The study refers to the measurement topics developed by Cordero [35], which are widely used. This part of the questionnaire was to be completed by the enterprise's middle and senior managers or its subsidiary managers. Representative items include the following: "In recent years, the company has speeded up product innovation" and "Technology innovation occurs through the introduction of new products and technology." The Cronbach's α of this part is 0.856.

The independent variable of this study is AI. We optimized and localized a scale based on the vital information extracted from a survey of large Chinese enterprises [36]. The questionnaire was also completed by the enterprise's middle and senior managers and subsidiary enterprise personnel. Representative items include "We will respond to changes in market demand quickly" and "We will respond to changes in market demand faster than our major competitors". The Cronbach's α of this part is 0.826.

The mediator is KS. We selected useful items from Hoof and Ridder, and collected self-evaluations by department managers [37]. Representative items include "When I learn new knowledge, I teach it to other colleagues"; "If I receive information, I share it with my colleagues"; "I teach my skills to other colleagues without reservation"; "When I do not understand something at work, I ask my colleagues; they teach me". The Cronbach's α of this part is 0.872.

The moderator is SL. In this part, we again rely on the widely used scale [36]. All managers were required to complete the questionnaire. Representative items include the following: "We find some of our strategies actually do not work"; "We can summarize the reasons why our strategies failed"; "We can learn from our mistakes and make fewer mistakes"; "We regularly change our strategies and act on them". The Cronbach's α of this part is 0.850.

Finally, the study's control variables are education level (C1), industry (C2), nature (C3), years (C4), whether the enterprise is a high-tech enterprise (C5), and whether the enterprise has received strategic guidance from a third-party consulting company (C6).

## Common method biases test

To avoid the potential impact of common method bias on the research results, this study reduced the impact to a certain extent through questionnaire pairing surveys, anonymous answers, design reverse questions, and other measures. Simultaneously, following Podsakoff *et al.*, the study primarily uses the Harman single-factor method to test for common method deviation [38]. The Harman single-factor method is an exploratory factor analysis without rotation for each latent variable item. If an independent factor is separated or the first factor's variance interpretation rate is more than 50%, a serious common method deviation is indicated [39]. In this study, no independent common factor was separated, and the variance interpretation rate of the first factor was 47.11%. Therefore, there was no serious common method bias in the study.

Table 3. Reliability, convergent validity, and discriminant validity.

| Construct | Factor load | Composite Reliability | Convergent Validity | | Discriminant validity | | | |
|---|---|---|---|---|---|---|---|---|
| | | CR | AVE | Cronbach's α | AI | KS | IB | SL |
| AI | 0.770–0.922 | 0.837 | 0.721 | 0.826 | **0.849** | | | |
| KS | 0.690–0.925 | 0.876 | 0.642 | 0.872 | 0.400 | **0.801** | | |
| IB | 0.668–0.831 | 0.859 | 0.604 | 0.856 | 0.903 | 0.540 | **0.777** | |
| SL | 0.567–0.803 | 0.853 | 0.541 | 0.850 | 0.785 | 0.471 | 0.699 | **0.736** |

Note: The main diagonal number is AVE's square root value, which is displayed in bold.

## Analysis and results

### Reliability and validity

First, we used Mplus 8.0 to analyze the reliability and validity of the questionnaire. Table 3 presents the load interval, combination reliability (CR), convergence validity (AVE), Cronbach's α, and discriminant validity test results for each latent variable factor. Regarding the reliability test, first, the combined reliability CR of the variables of AI, IB, SL, and KS is greater than 0.8, which conforms to the recommended standard of Fornell and Larcker (CR> 0.6), and the reliability test results are good [40]. Second, for each variable, Cronbach's α is greater than 0.8, which further indicates that the variables have good reliability. Regarding the validity test, the questionnaire's convergence validity was tested by measuring the mean-variance extraction (AVE). In Table 3, the load factors of each potential variable standardization factor were greater than 0.6, and the AVE values were greater than 0.6, which meets the recommendations of Fornell and Larcker on AVE standard value ($>$ 0.5) and indicates that the scale has good convergence validity [40]. Additionally, most of the square roots of AVE on the diagonal were greater than the direct correlation coefficients of the potential variables in the same row or column, indicating that the discriminant validity of the scales meets statistical requirements.

To further confirm discriminant validity, we used Mplus 8.0 software to perform confirmatory factor analysis. The results are shown in Table 4. First, all the four-factor model fitting indexes ($X^2$/df = 1.852; SRMR = 0.049; RMSEA = 0.053; CFI = 0.951; TLI = 0.938) meet the recommended standard, which indicates good differentiation between the several concepts involved in the study [41]. Second, compared with other nested models, such as the three-factor, two-factor, and single-factor models, the four-factor model has a higher fitting degree, which indicates that the latter model is the best.

Table 4. Construct validity analysis.

| Model | $X^2$ | DF | $X^2$/DF | Δ $X^2$ (ΔDF) | CFI | TLI | RMSEA | SRMR |
|---|---|---|---|---|---|---|---|---|
| Four-factor model (AI,SL,IB,KS) | 155.561 | 84 | 1.852 | - | 0.951 | 0.938 | 0.053 | 0.049 |
| Three-factor model (AI +KS,SL,IB) | 390.326 | 87 | 4.487 | 234.765 (3)*** | 0.791 | 0.748 | 0.107 | 0.136 |
| Three-factor model (AI,SL+IB,KS) | 251.236 | 87 | 2.888 | 95.675 (3)*** | 0.887 | 0.864 | 0.079 | 0.064 |
| Two-factor model (AI +KS,SL+IB) | 478.331 | 89 | 5.375 | 322.770 (5)*** | 0.732 | 0.684 | 0.120 | 0.143 |
| Two-factor model (AI +ORI,SL+KS) | 404.119 | 89 | 4.541 | 248.558 (5)*** | 0.783 | 0.744 | 0.108 | 0.091 |
| Single-factor model (AI +SL+IB+KS) | 483.902 | 90 | 5.377 | 328.341 (6)*** | 0.729 | 0.684 | 0.120 | 0.103 |

Note:

***refers to $p<0.001$;

**Table 5. Mean value, standard deviation, and correlation coefficient of variables.**

|     | Mean | SE   | 1        | 2        | 3        | 4       | 5       | 6      | 7       | 8       | 9       | 10      |
|-----|------|------|----------|----------|----------|---------|---------|--------|---------|---------|---------|---------|
| C1  | 4.06 | 0.76 | -        |          |          |         |         |        |         |         |         |         |
| C2  | 5.91 | 3.26 | -0.159** | -        |          |         |         |        |         |         |         |         |
| C3  | 3.68 | 0.72 | -0.050   | -0.11    | -        |         |         |        |         |         |         |         |
| C4  | 3.25 | 1.99 | -0.149** | 0.398**  | -0.178** | -       |         |        |         |         |         |         |
| C5  | 1.75 | 0.43 | -0.070   | 0.060    | -0.090   | 0.202** | -       |        |         |         |         |         |
| C6  | 1.57 | 0.50 | -0.159** | 0.240**  | -0.119*  | 0.362** | 0.316** | -      |         |         |         |         |
| AI  | 2.53 | 1.37 | 0.257**  | 0.040    | -0.040   | 0.060   | 0.110   | 0.000  | (0.826) |         |         |         |
| SL  | 2.62 | 1.14 | 0.188**  | -0.010   | -0.050   | -0.010  | 0.020   | -0.080 | 0.671** | (0.850) |         |         |
| IB  | 2.43 | 1.18 | 0.225**  | -0.020   | -0.080   | 0.050   | 0.126*  | 0.060  | 0.759** | 0.620** | (0.856) |         |
| KS  | 1.87 | 1.02 | 0.110    | -0.060   | -0.163** | 0.100   | 0.060   | -0.050 | 0.369** | 0.441** | 0.493** | (0.872) |

Note:

***refers to p<0.001;

** refers p<0.05;

* refers p<0.01;

In brackets is the Cronbach's value; C1-C6 refers to six control variables respectively.

Table 5 presents the mean, standard deviation, correlation coefficient, and square root of each variable's AVE value. AI was positively correlated with IB ($\beta = 0.759$, $p<0.05$), SL ($\beta = 0.671$, $p<0.05$), KS ($\beta = 0.369$, $p<0.05$), and KS ($\beta = 0.493$, $p<0.05$). Therefore, hypotheses H1, H2, and H3 were preliminarily verified.

## Main effect analysis

To test the main effect of the model, a series of structural equation models were constructed using Mplus8.0 software for hypothesis testing. The results are shown in Table 6. Here, M1 tests the impact of an agile market on IB ($X^2/df = 3.362$, CFI = 0.954, TLI = 0.913, RMSEA = 0.088, SRMR = 0.046), M2 tests the impact of AI on KS ($X^2/df = 2.033$, CFI = 0.983, TLI = 0.968, RMSEA = 0.058, SRMR = 0.036), and M3 tests the impact of KS on IB ($X^2/df = 1.234$, CFI = 0.994, TLI = 0.991, RMSEA = 0.028, SRMR = 0.039). The overall goodness of fit of the model is acceptable.

**Table 6. Results of main effect analysis.**

| Path and Model        |                      | M1       | M2       | M3       |
|-----------------------|----------------------|----------|----------|----------|
|                       |                      | AI→IB    | AI→KS    | KS→IB    |
| Path Coefficient      | AI→KS                |          | 0.399*** |          |
|                       | KS→IB                |          |          | 0.541*** |
|                       | AI→IB                | 0.900*** |          |          |
| Model Fitting Index   | $\chi^2/df$          | 3.362    | 2.033    | 1.234    |
|                       | CFI                  | 0.954    | 0.983    | 0.994    |
|                       | TLI                  | 0.913    | 0.968    | 0.991    |
|                       | RMSEA                | 0.088    | 0.058    | 0.028    |
|                       | SRMR                 | 0.046    | 0.036    | 0.039    |

Note:

***refers to $p<0.001$.

According to the results presented in Table 6, the effect path coefficient of AI on IB is 0.900 (P < 0.001), which indicates that an agile market response has a positive impact on organizational IB. Hypothesis H1 is further verified. The path coefficient of the effect of AI on KS was 0.399 (P < 0.001), which indicates that AI could promote KS behavior among department members. Hypothesis H2 was further verified. The effect path coefficient of KS on IB is 0.541 (P < 0.001), which indicates that KS among members can actively promote the IB of organizations. Therefore, research hypothesis H3 is verified.

## Mediating effect of KS

To test whether KS plays a mediating role in the positive impact of AI on organizational IB, this study uses bootstrap technology to test the mediating effect of KS (the results are shown in Table 7). According to a bootstrap analysis of 1000 repeated samples, KS's mediating effect (β = 0.079, SE = 0.025) is significant in a non-zero confidence interval (bias correction 95% CI is between 0.026–0.142; percentile 95% CI is between 0.029–0.145). These results support hypothesis H4.

## Moderating role of SL

This study adopts the hierarchical regression method to test SL's moderating effect on the relationship between AI and IB and the relationship between AI and KS. The results are shown in Table 8. To avoid the adverse effects of multicollinearity, we first centralized the latent variables and obtained the interaction terms. As presented in Table 8, the standardized cross-term coefficient of the moderating effect reveals that SL plays a negative moderating role in the positive effect relationship between AI and IB (β = - 0.088, P < 0.001), indicating that SL intensity plays an inhibitory role in IB promotion by top manager AI. Hypothesis H5a is verified. We believe that this result is universal in the context of Chinese culture. This is like a person has sufficient strategic knowledge, then changes in the external market arouse his vigilance, but in innovation may be more cautious. On the contrary, if a person is relatively lacking in strategic knowledge, the ups and downs in the field will cause him to be alert, and it is likely that he will immediately take innovative activities to deal with such changes in external conditions. To show SL's moderating effect on AI and IB's relationship intuitively, we draw a map of the SL moderating effect based on reference [42]. As shown in Fig 2, the influence curve of AI on IB is relatively flat at a high SL level and steeper at a low SL level. SL negatively moderates the positive impact of AI on IB.

**Table 7. Result of the mediating effect.**

| Effect | Standardization coefficient | SE. | Z-values | Bootstrapping | | | |
|---|---|---|---|---|---|---|---|
| | | | | Bias-Corrected 95% CI | | Percentile 95% CI | |
| | | | | Lower | Upper | Lower | Upper |
| Total effect effect | 0.900*** | 0.028 | 32.153 | 0.825 | 0.964 | 0.813 | 0.957 |
| Indirect effect effect | 0.079** | 0.025 | 3.211 | 0.026 | 0.142 | 0.029 | 0.145 |
| Direct effect | 0.820*** | 0.042 | 19.547 | 0.707 | 0.928 | 0.687 | 0.919 |

Note:

*** p<0.001;

** p<0.05;

* p<0.01;

Bootstrap = 1000.

**Table 8. Result of moderating effect.**

| steps | Variables and models | | IB | | | KS | | |
|---|---|---|---|---|---|---|---|---|
| | | | **M4** | **M5** | **M6** | **M7** | **M8** | **M9** |
| First step | control variables | Education | 0.373*** | 0.051 | 0.039 | 0.125 | -0.008 | -0.013 |
| | | industry | -0.007 | -0.023 | -0.023 | -0.031 | -0.037 | -0.037 |
| | | Working years | -0.069 | -0.05 | -0.043 | -0.212 | -0.191 | -0.188 |
| | | Enterprise properties | 0.025 | 0.001 | 0.007 | 0.079 | 0.071 | 0.073 |
| | | Is it a high-tech enterprise | 0.319 | 0.085 | 0.097 | 0.144 | 0.064 | 0.069 |
| | | Does it accept strategic guidance | 0.100 | 0.187 | 0.192 | -0.221 | -0.15 | -0.148 |
| Second step | Path a | independent variable: AI | | 0.523*** | 0.541*** | | | |
| | | moderator: SL | | 0.217*** | 0.221*** | | | |
| | Path b | independent variable: AI | | | | | 0.096 | 0.103 |
| | | moderator: SL | | | | | 0.307*** | 0.309*** |
| Third step | Moderating effects a | AI SL | | | -0.088*** | | | |
| | Moderating effects b | AI SL | | | | | | -0.036 |
| | | $F$ | 4.134 | 57.950 | 54.209 | 3.458 | 12.105 | 10.887 |
| | | $R^2$ | 0.077 | 0.61 | 0.623 | 0.065 | 0.247 | 0.249 |
| | | $\Delta R^2$ | 0.058 | 0.600 | 0.612 | 0.046 | 0.226 | 0.226 |

Note:

***refer to $p < 0.001$.

We also examine the standardized cross coefficient of the moderating effect b-path. The results show no significant moderating relationship between SL and KS ($\beta = -0.036$, $P > 0.05$), and hypothesis H5b is not verified. The study also verifies the moderated mediating effect (moderating effect of path c) of SL. After adding a standard deviation to and subtracting a standard deviation from SL, we used the bootstrapping method to test the mediating effect of different SL levels (below one standard deviation and higher than one standard deviation). It was found that the indirect effect, direct effect, and total effect of KS did not change significantly. Therefore, it is assumed that H5c is not verified. The test results for H5a, H5b, and H5c reveal that the AI of top managers promotes the IB of enterprises and that the higher the level of SL is, the more restrained the IB. In contrast, the lower that the level of SL is, the more pronounced the IB. Additionally, SL cannot affect the IB of enterprises through the moderating effect of KS.

## Discussion and conclusions

Currently, the world's political and economic pattern is changing significantly. At the enterprise strategic level, top manager AI with respect to the market has become crucial for enterprise survival and development. Similarly, only the top manager's market intuition is transformed into enterprise IB, and output (product/service) can form an enterprise competitive advantage. In this regard, scholars have undertaken research from the perspectives of business intuition [43], strategic intuition [44], and market sensitivity [45], but the results of these studies fail to fully reveal the procedural mechanism from the rapid introduction of market information to the final transformation into enterprise IB. Given the importance of this problem, based on survey data of 305 paired samples of top and department managers, this study investigated the mechanism relationship between top manager AI and enterprise IB and examined the roles of KS and SL in this framework. Based on our data test results, it can be concluded as follows: First, as the main managers of enterprises, their keen sense of market will

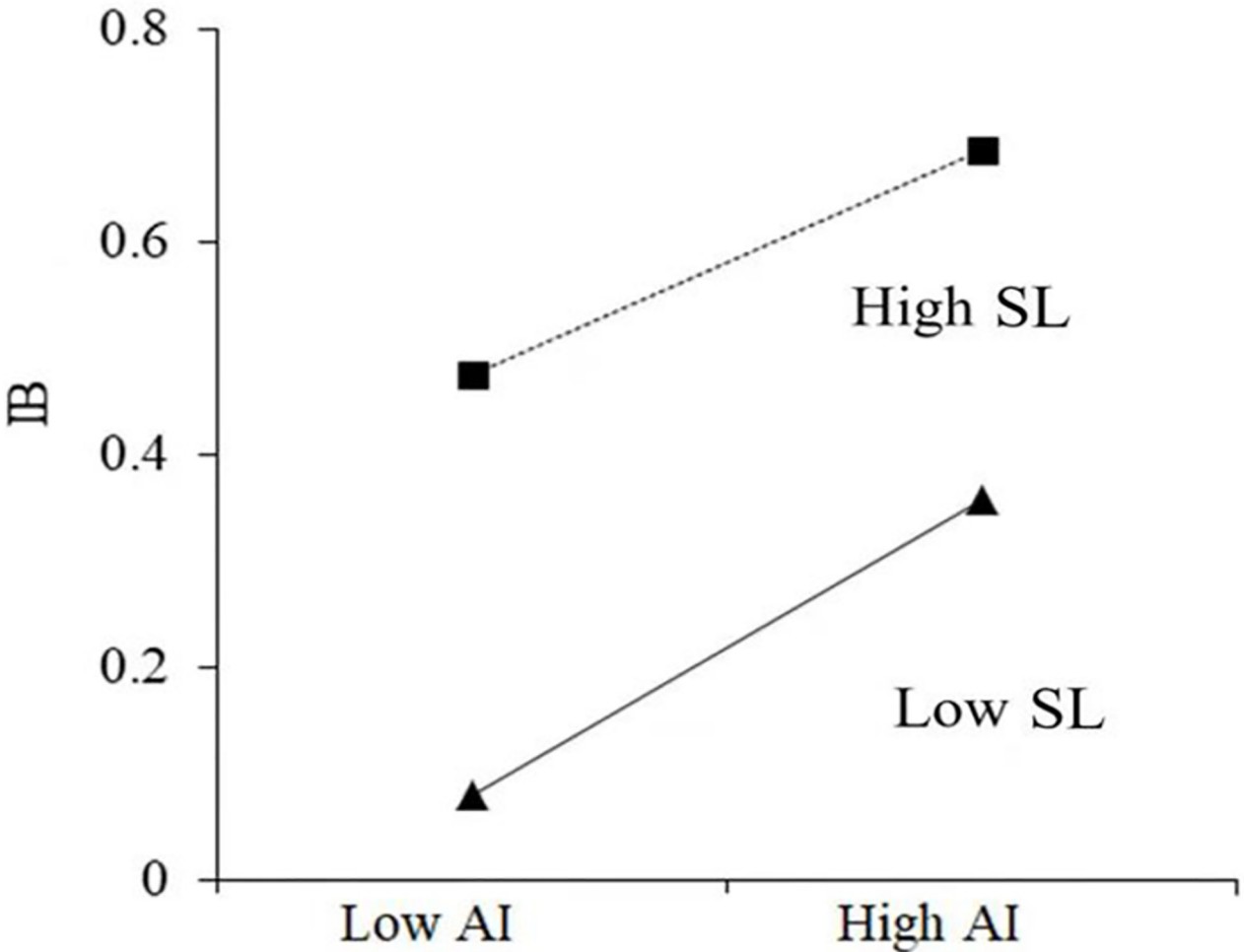

**Fig 2. The moderating effect of SL on the relationship between AI and organization innovation.**

directly promote the innovation behavior of the entire enterprise, which is consistent with the conclusions of many previous empirical studies. Second, the knowledge sharing mechanism within the enterprise plays a partial intermediary role in the mechanism of the managers' keen sense of the market and the enterprises' innovative behavior. In other words, the more perfect the knowledge sharing mechanism is within the enterprise, the more of managers' keen sense of smell will be transformed into the innovative behavior of the enterprise, and vice versa. This conclusion is rarely mentioned in previous studies. Third, the deeper the organizational strategy learning, the less the manager's keen perception of market changes will be transformed into corporate innovation behavior. This may be related to a person's personality, because the more knowledge accumulated, the more mature and cautious. This conclusion is quite different from previous studies.

## Contributions

Against the background of the rapid increase in resistance to economic globalization and an economic slowdown, top manager AI with respect to the business environment is becoming critical for enterprise survival and development. However, there is no consensus regarding how top management AI affects enterprise IB and which factors promote or inhibit this

process. Therefore, based on data collected from local enterprises, this study developed a new approach to investigate the effect mechanism of top manager AI and examined the influencing factors of enterprise IB. First, KS's mediating effect was investigated because it was found that top manager AI must be transformed into enterprise IB through the KS of the organization's entire membership. The study also focused on the moderating effect of SL. Unlike most research, this study found that at a higher level of SL the effect of top manager AI on enterprise IB is weaker than that at a lower SL level. Therefore, this study demonstrates that KS in fact promotes cognitive decision-making behavior. This study also emphasizes that SL's inhibitory effect should not be ignored (i.e., to prevent an organization's blind innovation caused by the top manager's inspiration). The practical implication is that KS can effectively promote the transformation of top manager AI into organizational IB. However, when SL becomes an active awareness or thinking orientation at different levels of the organization so as to communicate, it can also inhibit the influence of top manager agile intuition on organizational IB or slow organizational IB, thus preventing the top manager innovation risk caused by individualism and eagerness for quick success. Although the discovered mechanism cannot prove whether a person uses a dual decision-making system, the study results support the existence of such systems (including middle- and high-level managers and organizational members) depending on organizational level. The study also responds to Dijksterhuis [14]: under adverse circumstances, the market intuition of top managers may promote an organization's IB, but this phenomenon does not mean that the organization can produce better innovation results. The main reason is that SL inhibition may reduce IB blindness. That is, if the decision-making quality leads to innovative results rather than innovative behavior, then the "slow" (information-processing) decision-making system is more rational. This study finds that strategic learning level is not as high as the echelon theory suggests. A higher level of strategic learning enables managers to promote more innovative behaviors throughout the organization due to their sensitivity to market changes. On the contrary, the improvement of organizational strategy learning level inhibits managers' own judgments on market changes, which in turn inhibits the innovation activities of the entire organization. This conclusion may be more in line with the general laws of corporate strategic management under the Chinese cultural background, and expanding the new findings in this field.

Finally, this study enriches the findings of strategic choice theory in the context of China. For a long time, scholars have paid substantial attention to active agents with higher power (i.e., top managers), which makes the concept of "asymmetry" of corporate strategic choice at different organizational levels root deeply in the hearts of the people. Although this study does not deny this understanding, it emphasizes KS's bridging effect with respect to "asymmetry." That is, the market intuition of top managers facilitates forming a consensus of the entire organization through systematic knowledge dissemination, sharing, and interaction and can promote organization IB. For example, in certain rapidly developing industries, enterprises tend to adopt brainstorming to fully communicate and share views with employees at different levels and thus to promote the emergence of new ideas and technologies [46,47]. This result also confirms the view of reference [48].

## Managerial implications

In the long run, innovation is the essence of an enterprise [49]. Because of the top manager's status and power, his or her AI has become an essential factor in enterprise entrepreneurial behavior. This study's conclusion should encourage enterprises to attach importance to AI regarding the market and emphasizes the significance of strengthening KS and SL. The specific management recommendations are as follows.

First, against the background of significant changes in the market structure, enterprise top managers should pay closer attention to the accurate perception and prediction of the market and seek to consider their AI at all times. For example, Tencent, TikTok, and other enterprises were able to discover a large opportunity in the remote office, front door, game, live broadcast market under the epidemic situation, quickly responded to this market demand, and realized a new round of peak revenue and market value.

In addition, efficient knowledge dissemination and sharing mechanisms determine enterprise quality and efficiency with respect to transforming market intuition to IB. A talented businessperson must quickly capture critical market information and then predict market prospects. However, this type of anticipation cannot be fully understood and digested by organization members, which affects organizational results. To overcome this problem, it is necessary to systematically process the intuitive anticipation of top managers and encourage interaction with respect to knowledge in a manner that can be read, understood, and shared by all organization members. However, it is risky to base strategy, decision-making, and other important tasks entirely on the intuition and inspiration of entrepreneurs. Enterprise top managers should fully trust and affirm the useful contributions of all organization members, spread and ensure interaction regarding the knowledge of entrepreneurs in the organization, form an efficient work sequence, and thus realize enterprise goals.

Finally, SL should be multilevel organizational learning. There are many ways for top managers to learn strategies, such as 1) hiring strategic consultants to assess industry and competitor information for decision-making and 2) organizing peers to communicate to form an atmosphere of mutual learning. Regarding the SL of members, such as encouraging middle- and high-level and even front-line employees to undertake further study and compete regarding strategic knowledge, SL encourages the entire staff's understanding of strategy. This study finds that SL's value lies in the rational moderation-inhibition effect between top manager AI and enterprise IB. For example, in the absence of an SL atmosphere and management, entrepreneurs quickly pass their own "flash" intuitions to the technology and product development teams. This approach is likely to result in blind and ineffective innovation results. In traditional enterprises with an intense SL atmosphere, the inspiration of top managers is restrained by SL level to avoid a mismatch between IB and innovation results.

## Limitations and future research direction

There are several limitations to this study. First, because of limited funding and research conditions, the study did not conduct a large-scale survey of all industries and enterprises in China. In addition, the primary respondents were enterprises of a particular scale and enterprises familiar with the research team, which may have caused us to neglect investigating other industries and enterprises. Regarding market AI, enterprises primarily rely on the rational judgment of enterprise managers while ignoring the intuitive observations of such managers. Future research should further investigate the top manager AI, entrepreneurial intuition, and business intuition of organizations in a multidimensional manner. Finally, as a fundamental theoretical discovery, the SL mechanism in the context of organizational intuition-decision-behavior should receive more research attention.

## Supporting information

**S1 Data.**
(ZIP)

## Author Contributions

**Conceptualization:** Qing Zhao.

**Data curation:** Mei Yu, Shu Shang, Yuqi Zhu.

**Formal analysis:** Qing Zhao, Shu Shang, YanPing Xie.

**Funding acquisition:** Qing Zhao.

**Methodology:** Langang Feng, Yuzhu Meng.

**Software:** Langang Feng, Yuzhu Meng.

**Supervision:** Jing Li.

**Validation:** Yuqi Zhu, Jing Li.

**Visualization:** Mei Yu.

**Writing – original draft:** Mei Yu.

**Writing – review & editing:** Hai Liu, Mei Yu, Shu Shang.

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
