## [Decision Letter · Decision Letter 0]

6 Oct 2021

PONE-D-21-09995The Impact of Agile Intuition on Innovation Behavior: Chinese Evidence and a New ProposalPLOS ONE

Dear Dr. Yu,

Thank you for submitting your manuscript to PLOS ONE. After careful consideration, we feel that it has merit but does not fully meet PLOS ONE’s publication criteria as it currently stands. Therefore, we invite you to submit a revised version of the manuscript that addresses the points raised during the review process. Both the Reviewers recommended minor revision. You are expected to go through each one of the addressed points and to come up with a more robust manuscript.

We look forward to receiving your revised manuscript.

Kind regards,

Alessandro Margherita

Academic Editor

PLOS ONE

Journal Requirements:

3. Please provide copies of all questionnaires, with English translations, used in your study as supporting information files."

PXB (EO) 14 Apr 21: 1. Thank you for including your ethics statement:  "N/A

In each industry, we selected 5-10 representatives who were familiar with the research.

Under consent, the administrative departments of the enterprise were entrusted with

issuing the questionnaire. Finally, the questionnaires of 47 enterprises were collected,

including manufacturing, trade, finance, and engineering.

5. Please provide additional details regarding participant consent. In the ethics statement in the Methods and online submission information, please ensure that you have specified what type you obtained (for instance, written or verbal, and if verbal, how it was documented and witnessed). If your study included minors, state whether you obtained consent from parents or guardians. If the need for consent was waived by the ethics committee, please include this information.

6. Please update your submission to use the PLOS LaTeX template. The template and more information on our requirements for LaTeX submissions can be found at http://journals.plos.org/plosone/s/latex.

7. In your Data Availability statement, you have not specified where the minimal data set underlying the results described in your manuscript can be found. PLOS defines a study's minimal data set as the underlying data used to reach the conclusions drawn in the manuscript and any additional data required to replicate the reported study findings in their entirety. All PLOS journals require that the minimal data set be made fully available. For more information about our data policy, please see http://journals.plos.org/plosone/s/data-availability.

8. Please include your full ethics statement in the ‘Methods’ section of your manuscript file. In your statement, please include the full name of the IRB or ethics committee who approved or waived your study, as well as whether or not you obtained informed written or verbal consent. If consent was waived for your study, please include this information in your statement as well.

Reviewers' comments:

Reviewer's Responses to Questions

**Comments to the Author**

1. Is the manuscript technically sound, and do the data support the conclusions?

Reviewer #1: Yes

Reviewer #2: Yes

2. Has the statistical analysis been performed appropriately and rigorously? 

Reviewer #1: Yes

Reviewer #2: Yes

3. Have the authors made all data underlying the findings in their manuscript fully available?

Reviewer #1: Yes

Reviewer #2: No

4. Is the manuscript presented in an intelligible fashion and written in standard English?

Reviewer #1: Yes

Reviewer #2: Yes

5. Review Comments to the Author

Reviewer #1: 1. The introduction part introduces the important influence of the Agile Intuition of top managers on the innovation behavior of enterprises. The research questions are derived from the problems and the status quo, with a rigorous logic and hierarchy. But Line 99, it is not very appropriate to directly give the influence of KS and SL on innovation behavior, without any foreshadowing and presentation of related concepts and researches. Here readers do not know what KS and SL are.

2. In the part of literature review and hypothesis, we can clearly understand the impact of AI on IB, as well as the possible mediating role of KS and moderating role of SL. However, for Hypothesis 5: SL moderates the positive effect of AI on IB, is there more support and literature to explain it?

3. In the section of sample and data collection, authors distribute and collect two sets of questionnaires to middle- and high-level managers and heads of subsidiaries to measure AI and SL, respectively. Than three sets of questionnaires were distributed and collected to measure SL, KS, and IB. So, how is the SL questionnaire measured, and how many times does it measure? Does the group of subjects change? Please state it more clearly.

4. Following question 3, the research object is the AI of top managers, involves SL of top managers and KS with employees. We see Line 235: there are 112 middle and senior managers and 356 subsidiary managers. Although there are only 305 valid samples, but in Line 240 and Table 1: 74.43% are ordinary staff, and 25.57% were middle and senior managers. Where do ordinary employees come from? Please explain and standardize the subject group in detail.

5. Regarding the phenomenon in Figure 2, why does this result occur, is it related to the Chinese cultural background or is it universal? Please give a detailed explanation and description.

6. This study conducts a relatively sufficient data analysis on the hypothesis, but there are still shortcomings in the discussion. For examples, data analysis supports Hypothesis 5, but how about its theoretical analysis and discussion? Authors need more discussion to reach the 3 conclusions in line 416-418. Please supplement more analyze and discuss the impact of the results of previous section from various aspects to support the conclusions of this article, not just restate the research results.

Reviewer #2: 1) Update literature on the analysed topics (more recent articles are of 2019);

2) clarify better the real research gap already in the Introduction (use extant literature to justify the research goals);

3) provide a much more extended discussion, especially in terms of the advancement respect to extant theory;

4) proofread the manuscript for English style and grammar.

6. PLOS authors have the option to publish the peer review history of their article (what does this mean?). If published, this will include your full peer review and any attached files.

Reviewer #1: No

Reviewer #2: No

---

## [Author Response · Author response to Decision Letter 0]

25 Dec 2021

Reviewer #1: 1. The introduction part introduces the important influence of the Agile Intuition of top managers on the innovation behavior of enterprises. The research questions are derived from the problems and the status quo, with a rigorous logic and hierarchy. But Line 99, it is not very appropriate to directly give the influence of KS and SL on innovation behavior, without any foreshadowing and presentation of related concepts and researches. Here readers do not know what KS and SL are.

[Respond] Thanks for your suggestions, we added the following presentations in line 93-101:

We believe that based on the high-echelons theory and leader-member exchange theory, most studies can prove that when managers become more sensitive to market changes, it can be translated into innovation activities throughout the enterprise so that they can gain more advantage in competition. Nevertheless, it is not yet clear how leaders communicate and share information and knowledge with corporate members during the transformation process, as well as the role of these important factors in the corporate strategy learning and knowledge accumulation of the entire organization. In addition, currently few related studies were found taking into account the mechanism of knowledge sharing and strategic learning.

2. In the part of literature review and hypothesis, we can clearly understand the impact of AI on IB, as well as the possible mediating role of KS and moderating role of SL. However, for Hypothesis 5: SL moderates the positive effect of AI on IB, is there more support and literature to explain it?

[Respond] The current literature has not discussed the regulatory effect of SL on AI and IB, so we make a bold hypothesis and try to verify whether this regulatory effect exists through the current survey. In order to avoid missing our discussion on the regulatory role of SL, we also divide the regulatory role of SL into three dimensions for testing, which constitute hypotheses 5a, 5b and 5c. We are pleased that the regulatory effect hypothesis of SL is valid in 5a, while 5b and 5c are not valid.

3. In the section of sample and data collection, authors distribute and collect two sets of questionnaires to middle- and high-level managers and heads of subsidiaries to measure AI and SL, respectively. Than three sets of questionnaires were distributed and collected to measure SL, KS, and IB. So, how is the SL questionnaire measured, and how many times does it measure? Does the group of subjects change? Please state it more clearly.

[Respond] This proposal is very important and necessary. In this regard, we further elaborate on the issue and collection of questionnaires. In line 240-251, we add the following description. 

We conducted a total of two questionnaire surveys. Two sets of questionnaires were issued for the first time to measure AI and SL. The subjects included 112 senior executives from the parent company and 356 senior executives from the subsidiary. 936 copies distributed and 766 valid ones returned, with an effective recovery rate of 82.3%. The first test is SL. Three sets of questionnaires were issued for the second time to measure SL, KS and IB. The subjects were 468 non-leaders from the 47 enterprises, a total of 1404 copies distributed and 852 valid ones returned, with an effective recovery rate of 60.7%. As for the SL in both of the two surveys, we used exactly the same questionnaire in terms of participants, with slightly different expression of the questions, allowing the interviewees to accurately identify these questions. Totally 1618 copies of all the five sets of valid questionnaires were matched, and finally 305 data tables were obtained, which can completely cover the five latent variables.

4. Following question 3, the research object is the AI of top managers, involves SL of top managers and KS with employees. We see Line 235: there are 112 middle and senior managers and 356 subsidiary managers. Although there are only 305 valid samples, but in Line 240 and Table 1: 74.43% are ordinary staff, and 25.57% were middle and senior managers. Where do ordinary employees come from? Please explain and standardize the subject group in detail.

[Respond] Here this article has an inaccurate expression, in response. The middle-level and high-level managers mentioned in this paper include both the senior managers of the parent company and the senior managers of the subsidiaries. Our information statisticians made a mistake in calculation, regarding the senior managers of subsidiaries as ordinary employees. In fact, we surveyed a total of 112 top managers of the parent company, 356 top managers of the subsidiary, a total of 468 people, a total of 383 people return visit data is effective. We also surveyed 468 non-leaders, but only 284 provided valid questionnaires. Thus, 383 of the respondents were senior managers, accounting for 57.4%, and 284 were non-managers, accounting for 42.6 %. We have corrected these data in line 240-251.

5. Regarding the phenomenon in Figure 2, why does this result occur, is it related to the Chinese cultural background or is it universal? Please give a detailed explanation and description.

[Respond] The phenomenon in Figure 2 reflects that the degree of strategic learning inhibits the promoting effect of market sensitivity on innovation behavior. We believe that this result is universal in the context of Chinese culture. This is like a person to learn more strategic knowledge, then changes in the external market caused his vigilance, but in innovation may be more cautious. On the contrary, if a person’ s strategic knowledge is relatively lacking, the ups and downs in the field will cause him to be alert, and it is likely to immediately make him take innovative activities to cope with such changes in external conditions.It has been supplemented at lines 396-401.

6. This study conducts a relatively sufficient data analysis on the hypothesis, but there are still shortcomings in the discussion. For examples, data analysis supports Hypothesis 5, but how about its theoretical analysis and discussion? Authors need more discussion to reach the 3 conclusions in line 416-418. Please supplement more analyze and discuss the impact of the results of previous section from various aspects to support the conclusions of this article, not just restate the research results.

[Respond] According to the opinions of the reviewers, we have modified the relevant conclusions supporting hypothesis 5, modifications at lines 436-447，as follows. 

First, as the main managers of enterprises, their keen sense of market will directly promote the innovation behavior of the entire enterprise, which is consistent with the conclusions of many previous empirical studies. Second, the knowledge sharing mechanism within the enterprise plays a partial intermediary role in the mechanism of the managers’ keen sense of the market and the enterprises’ innovative behavior. In other words, the more perfect the knowledge sharing mechanism is within the enterprise, the more of managers’ keen sense of smell will be transformed into the innovative behavior of the enterprise, and vice versa. This conclusion is rarely mentioned in previous studies. Third, the deeper the organizational strategy learning, the less the manager’s keen perception of market changes will be transformed into corporate innovation behavior. This may be related to a person's personality, because the more knowledge accumulated, the more mature and cautious. This conclusion is quite different from previous studies.

Reviewer #2: 1) Update literature on the analysed topics (more recent articles are of 2019);

[Respond] We have consulted the latest literature and supported some viewpoints in the text. There are two articles updated to 2021 in the references, the relevant literature was added to line 495 and line 498, respectively.

2) clarify better the real research gap already in the Introduction (use extant literature to justify the research goals);

[Respond] According to the reviewers’ opinions, we further clarify the gaps in the study in the introduction. See lines 93-101 for details. 

We believe that based on the high-echelons theory and leader-member exchange theory, most studies can prove that when managers become more sensitive to market changes, it can be translated into innovation activities throughout the enterprise so that they can gain more advantage in competition. Nevertheless, it is not yet clear how leaders communicate and share information and knowledge with corporate members during the transformation process, as well as the role of these important factors in the corporate strategy learning and knowledge accumulation of the entire organization. In addition, currently few related studies were found taking into account the mechanism of knowledge sharing and strategic learning.

3) provide a much more extended discussion, especially in terms of the advancement respect to extant theory;

[Respond] We further discuss the advancement of existing theories in the contribution section. See lines 476-484 for details. This study finds that SL is not as high as the echelons theory shows. A higher level of strategic learning enables managers to promote more innovative behaviors of the entire organization due to their sensitivity to market changes. On the contrary, because of the improvement of organizational strategic learning level, it inhibits the managers’ own judgment of market changes, and then more than the whole organization innovation activities. This conclusion may be more in line with the general law of corporate strategic management in Chinese cultural background, expanding the new findings in this field.

4) proofread the manuscript for English style and grammar.

[Respond] We proofread the manuscript and revised the English style and grammar again.

---

## [Editor Report · Decision Letter 1]

27 Dec 2021

The Impact of Agile Intuition on Innovation Behavior: Chinese Evidence and a New Proposal

PONE-D-21-09995R1

Dear Dr. Yu,

We’re pleased to inform you that your manuscript has been judged scientifically suitable for publication and will be formally accepted for publication once it meets all outstanding technical requirements.

Kind regards,

Alessandro Margherita

Academic Editor

PLOS ONE
---

## [Editor Report · Acceptance letter]

20 Apr 2022

PONE-D-21-09995R1 

Impact of Agile Intuition on Innovation Behavior: Chinese Evidence and a New Proposal 

Dear Dr. Yu:

I'm pleased to inform you that your manuscript has been deemed suitable for publication in PLOS ONE. Congratulations! Your manuscript is now with our production department. 

Kind regards, 

on behalf of

Dr. Alessandro Margherita 

Academic Editor

PLOS ONE